# Research Regarding the Energy Recovery from Municipal Solid Waste in Maramures County Using Incineration

**Miorita Ungureanu [1], Juhasz Jozsef [1], Valeria Mirela Brezoczki [1], Peter Monka [2] and Nicolae Stelian Ungureanu [1,\***]

[1] North University Centre at Baia Mare, Faculty of Engineering, Technical University of Cluj-Napoca, V. Babes St. 62, 430083 Baia Mare, Romania; miorita.ungureanu@cunbm.utcluj.ro (M.U.); jozsef.juhasz@cunbm.utcluj.ro (J.J.); valeria.brezoczki@cunbm.utcluj.ro (V.M.B.)

[2] Faculty of Production Technologies with the Seat in Prešov, Technical University of Kosice, Štúrova 31, 080 01 Prešov, Slovakia; peter.pavol.monka@tuke.sk

\* Correspondence: nicolae.ungureanu@cunbm.utcluj.ro; Tel.: +40-745-298-976

**Abstract:** This paper presents a part of the study referring to exploring Energy Recovery from Municipal Solid Waste in Maramures County. In order to analyze the possibility of energetic recovery of municipal solid waste (MSW), data referring to the management system of MSW from Maramures county were cumulated and processed in a first stage in order to estimate the quantity of municipal solid waste and its composition, which might be recovered energetically. In the next stage, samples of municipal solid waste were collected from landfills, which were submitted to specific processing and analyses. The experimental data were processed and in the end the energy potential of municipal solid waste from Maramures county was found. This study will help stakeholders and those involved in waste management to assess the possibility of energy recovery. The analysis of the study concluded that municipal solid waste in Maramures County is a potential source of renewable energy.

**Keywords:** municipal solid waste; calorific value; energy potential





## 1. Introduction

The sustainable management of the waste is essential for the society. In the prioritization of the waste management, the following hierarchy is known: Prevention, reuse, recycling, recovery, and disposal [1].

The management of a landfill is disadvantageous for the environment and involves covering some areas of ground, and that is why another essential sector, the energy sector, is considered a perfect alternative, as municipal solid waste (MSW) is classified as a source of energy [2].

The energy recovery from municipal solid waste might play an important role in the transition to a circular economy, on condition that the processes of prevention, reuse and recycling should be priority in the systems of waste management [3,4].

The solid municipal waste has an important calorific value and burning the waste in an incinerator may be used in order to generate electric energy or heat meant to heat the population [5].

The most common thermal treatment process for MSW is incineration (generally without being treated before), this method is considered to be the most reliable and economical form of energy recovery MSW [6,7].

The waste incineration using the last generation technology usually requires a minimum temperature of 850 °C for a dwelling period in the heating chamber of 2 s and a good turbulence, with the minimum content of oxygen specific for the systems (for example at least 3% excess of oxygen in the free gas after the incineration in a system with fluidized bed) [8].

An important advantage of waste incineration is the fact that it represents a rapid waste treatment method, with very large quantities being destroyed in a relatively short span of time. Modern incinerators reduce the volume of the waste by 95–96 percent, depending on the composition of the waste, on the degree of recovering certain materials such as metals, glass, and the recyclation degree [5]. The reduction of waste volume by incineration leads to the reduction of the area necessary for storing and using them in different purposes. Waste incineration permits the destruction of organic pollutants and substances [8]. Another advantage is the possibility to use the ashes and the solid waste in the road construction and cement industries [6].

The process of thermal treatment also reduces at zero the danger of infestation of the ground water by possible infiltrations of the leachate resulting in deposits, and reduces the methane emissions by the abolished landfills [4].

Another important advantage is the fact that the energy recovery leads to the reduction in the consumption of conventional fuels of heat or electricity.

Landfills have been shown to be a source of greenhouse gas (GHG) and carbon dioxide ($CO_2$) emissions [9,10]. In this context one advantage to be highlighted in the case of incineration is the total elimination of methane emissions and the quantity of CO2 emissions of MSW incineration highly depends on the waste composition and plant technology [8]. Landfills also emit organic compounds and inorganic compounds that cause odors and health problems for local citizens [9]. A systemic approach of sustainable action is essential in ensuring a favorable housing climate for citizens, in maintaining habitats and protected areas [11].

A disadvantage of waste incineration is a series of emissions resulting from their combustion. But according to the literature, these emissions can be reduced so that they fall within the limits allowed by the laws and regulations on industrial emissions [12,13].

Combustion of MSW in the incinerator results in flue gases, especially CO, $CO_2$, $H_2O$, NOx, and, if applicable, $SO_2$. To a lesser extent, acid gases such as HCl and HF are also produced, and, last but not least, heavy metals and macromolecules with high stability and higher molecular weight (dioxins, Furan's and PCB's) [7]. Under certain conditions dioxins, furans, and similar gaseous components are only destroyed; the rate of organic molecules' destruction depends on the high temperature inside the furnace and the residence time of combustion gases in the incinerator [8].

The emissions resulting from MSW incineration are greatly reduced by combustion technologies. But apart from these measures, modern incinerators are equipped with filtering systems for the resulting emissions. So, the modern incinerator is an efficient combustion system, which produces energy and reduces waste to an inert residue with minimal pollution [14]. But the cost of an emission minimization technology for an incinerator can be up to 35% of the project cost [5].

To take into account the possibility of energy recovery by incineration, the amount of waste used for incineration must not be less than 50,000 tons per year [8], and the calorific value must be greater than 7 MJ/kg [15]. Energy efficiency for incineration facilities [16], life cycle energy assessment for incineration facilities, economic impact, and social impact assessment [17] are aspects that need to be taken into account.

Some problems can be encountered during management of solid wastes since they have a heterogeneous structure. For this reason, physical features of solid wastes, such as moisture content (MC), calorific value (heating) (HHV), and composition, should be well known for their management through suitable methods [11,18].

In order to evaluate the possibility of energy recovery of MSW by incineration, it is first necessary to analyze the waste generation method, and then the following characteristics of MSW: Composition, moisture content (MC), chemical characteristics, and calorific value (heating) [1,2].

In conclusion, we can highlight three advantages in the energy recovery from municipal solid waste: Environment protection, energy production from non-conventional sources, and hygiene reasons.

A systemic approach of sustainable action is essential in ensuring a favorable housing climate for citizens, in maintaining habitats and protected areas [18].

The main motivation of this research is to contribute to the reduction, and if possible, to the elimination of landfills and to obtaining energy, an essential element for the transition to a circular economy.

## 2. Materials and Methods

In the first part of our study, we will analyze waste generation in Maramures County, and later we will analyze the characteristics of municipal solid waste in terms of energy. In this context, our study begins from the premise that municipal solid waste is regarded as a chemical fuel and specific experiments and analyses are carried out in this context. A characterization of a chemical fuel involves. A characterization of a chemical fuel requires the knowledge, in addition to its physical state and origin, also of characteristics such as: Chemical composition, calorific value, and moisture [5]. Finally, the calculations performed in order to determine the energy potential of municipal solid waste in the county are presented.

### 2.1. Waste Generation in Maramures County

The amount of municipal solid waste and its composition undergo changes depending on the consumption habits of the population which are constantly changing [19]. The values and characteristics of municipal solid waste differ not only from one country to another, but also from one region to another, even from one neighborhood to another in the same city [20,21].

The research area, the Maramureș County, is located on the North of Romania. The chief town is the municipality of Baia Mare. The area of the county is of 6215 km². Maramureș County has more than 500,000 inhabitants. The county is made up of 2 municipalities: Baia Mare, Sighetul Marmatiei, 11 towns and 63 villages. Currently, all urban and rural localities in Maramureș County benefit from sanitation services.

Most of the waste collected in the county is disposed of by landfill in two landfills located in Satu Nou de Jos (Baia Mare) and Sighetu Marmației (Figure 1) with an area of about 22 ha and a capacity of 4.2 million cubic meters are managed by two private companies [22].

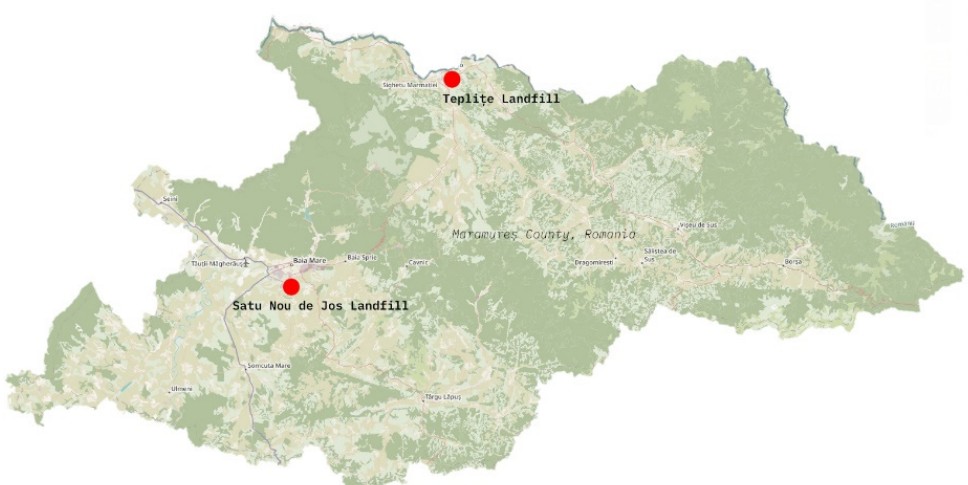

**Figure 1.** The landfills from Maramures County [23].

The total quantity of municipal solid waste generated in the year 2017 was of 86,382.3 tons of waste, and from this quantity 83,410 tons have been eliminated by removal [24]. In the year 2018 a quantity of municipal solid waste of 81,923.86 tons was generated, out of which 71,302 tons were deposited in landfills [25].

Our study considers the analysis of the energy potential of the amount of waste that is stored annually in the two landfills, in order to recover energy through incineration.

In order to determine the energy potential of municipal solid waste, we must know two main parameters: The calorific value of this waste and the amount of waste to be incinerated [2,26]. In addition to these two parameters, we have also determined other important characteristics in the study: The composition of the samples, the moisture of the samples and the elementary chemical composition of the samples.

### 2.2. The Estimated Municipal Solid Waste Quantity for Energy Recovery

In a first stage, based on the data provided by the accredited environmental agency in the county, the amount of waste available for incineration was determined, based on statistics from the last two years (2017 and 2018) and based on policies to increase recovery and recycling. Thus, based on the collected data, the total amount of municipal solid waste estimated for energy recovery is 53,325.87 tons per year and the distribution by waste categories is presented in Table 1. The amount available for incineration was obtained after deducting from the total amount of waste collected the part of recoverable and recyclable waste, glass, and inert waste from construction. This estimated annual amount of municipal solid waste for incineration and the chemical composition are shown in Table 1.

**Table 1.** Estimated annual amount of municipal solid waste (MSW) for incineration.

| Physical Composition of MSW | Quantity (t) | Percentage |
|---|---|---|
| Paper, cardboard | 5683.49 | 10.66 |
| Plastic | 5718.14 | 10.72 |
| Wood | 2571.99 | 4.82 |
| Organic matter | 30,794.78 | 57.75 |
| Textile | 3998.28 | 7.5 |
| Bulk waste | 4559.21 | 8.55 |
| Total | 53,325.87 | 100 |

The amount of waste generated per day to be incinerated is 146.1 t/day. In the case of incineration, facilities are justified above 100 t/day [6].

In conclusion, regarding the amount of waste generated per day, incineration is justified.

### 2.3. Collection and Preparation of Samples

Collecting waste samples for the study was the next important step. Obtaining a representative sample of one gram from a garbage truck full of waste is a very difficult operation, even if strict sampling and processing procedures are followed. Larger-scale instruments (1 kg) have been reported to determine the calorific value of waste, specifically designed for the analysis of municipal solid waste [5].

Samples were collected and prepared according to the study plan [27]. Due to the fact that the aim of the study is to determine the annual energy potential of municipal solid waste in Maramures County and to perform this calculation we refer to the total amount of waste presented in Section 2.2. The physical composition of samples was chosen to be consistent with the annual composition of waste. Under these conditions, based on the data collected regarding the percentage composition of MSW per year presented in Table 1, the samples were taken in compliance with the percentages. The municipal solid waste samples were collected from the two landfills in the county, Satu Nou de Jos (Figure 2a) and Teplițe (Figure 2b). At the location of Satu Nou de Jos there is also a sorting station. The samples were taken from fresh waste arrived in the landfill, it was put in bags on the categories of materials presented in Table 1, the bags were transported to the EnyMSW laboratory, where the sorting and weighing by categories was performed.

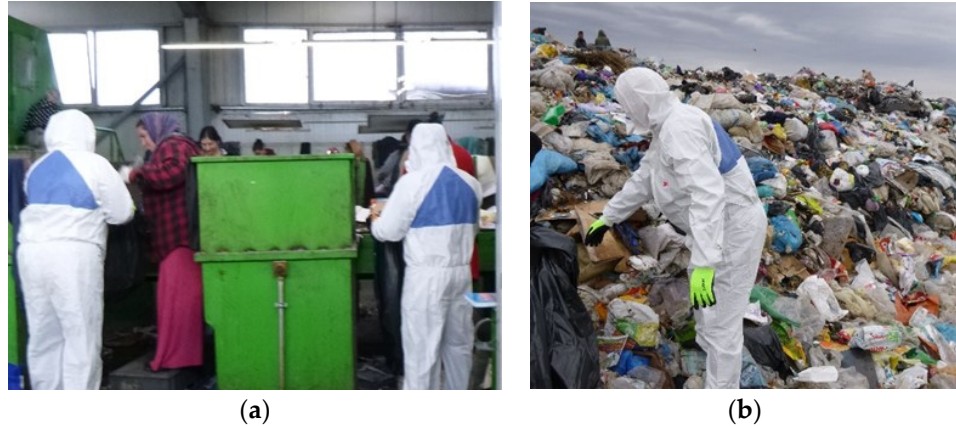

| (**a**) | (**b**) |

**Figure 2.** Samples collection: (**a**) From la Satu Nou de Jos; (**b**) from Teplite.

Each waste bag was marked with the date of collection, the place of collection, and the percentage composition (%) was classified into seven categories as follows:

- Organic matter in percentage of 57.75%;
- 10.72% plastic;
- 10.66% of paper and cardboard;
- 7.5% of textile materials;
- 4.82% of wood; and
- 8.5% bulk waste.

Thus, five samples, each 1 kg, were selected from the 2 landfills in Maramures County. The five samples were prepared and subjected to analysis and determination. After a first drying and the determination of the relative moisture, the samples were ground with Cutting Mill PULVERISETTE 15, the material resulting after grinding was sieved and homogenized (Figure 3).

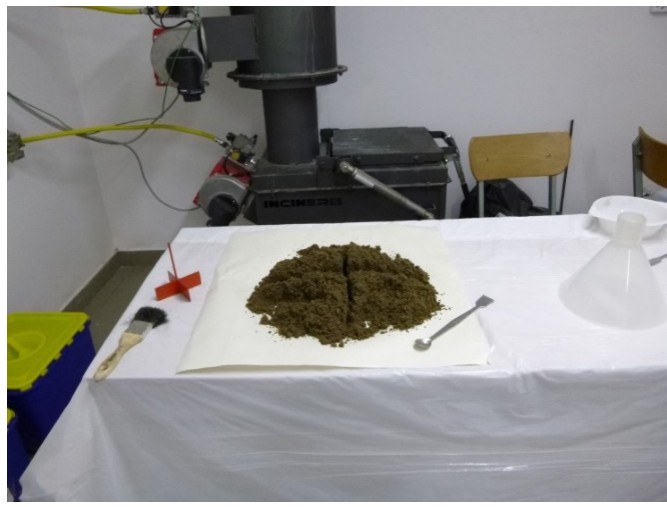

**Figure 3.** Ground sample.

*2.4. Moisture Content of the Municipal Solid Waste*

The moisture content was performed by laboratory drying of an analytical sample, by drying the samples, until constant mass was reached. The results are presented in Table 2. The relative moisture was determined before grinding the samples by drying in the open air at the temperature of 20 °C with a relative moisture of the air of about 50%. The hygroscopic moisture was determined by drying in a laboratory oven at 105 °C.

**Table 2.** Moisture content of the MSW.

| Sample Number | Relative Moisture Sample [%] | Hygroscopic Moisture [%] | Total Moisture [%] |
|---|---|---|---|
| Sample 1 | 3.69 | 40.01 | 42.22 |
| Sample 2 | 2.18 | 42.08 | 43.34 |
| Sample 3 | 3.97 | 41.21 | 43.54 |
| Sample 4 | 2.13 | 41.93 | 43.17 |
| Sample 5 | 2.78 | 40.84 | 42.48 |

The total moisture was calculated as follows [28]:

$$M_T = M_r + M_h(100 - M_r)/100 \tag{1}$$

where $M_h$ is hygroscopic moisture [%] $M_r$ and relative moisture [%].

*2.5. Chemical Composition of Municipal Solid Waste*

The chemical composition of municipal solid waste has most influence on the treatment method and recovery options [1,5]. In our study it is necessary to know the chemical composition of the samples to determine the net calorific value of the samples after obtaining experimentally a gross calorific value.

The chemical composition of municipal solid waste was determined by elementary chemical analysis in the Gas and Fuel Laboratory I.C.S.I. Rm. Vâlcea and resulted in the percentage content of carbon (C), hydrogen (H), sulfur (S), oxygen (O), and nitrogen (N) in the organic mass of the fuel for the samples. The chemical determination procedures for these elements are shown in Table 3 and the results of the determinations are shown in Table 4. The determinations were performed after the samples were dried.

**Table 3.** Chemical determination procedure.

| Name of Determination | Procedure |
|---|---|
| Carbon | ASTM D 5373-16PS-AGC-15 |
| Hydrogen | ASTM D 5373-16PS-AGC-15 |
| Nitrogen | ASTM D 5373-16PS-AGC-15 |
| Sulphur | PS-AGC-15 |
| Oxygen | Calculus |

**Table 4.** Chemical composition of the MSW sample.

| Sample No | Carbon [%] | Hydrogen [%] | Nitrogen [%] | Sulphur [%] | Oxygen [%] |
|---|---|---|---|---|---|
| Sample 1 | 7.25 ± 1.07 | 3.74 ± 0.09 | 0.46 ± 0.02 | S < LQ [1] | 17.75 |
| Sample 2 | 46.47 ± 1.33 | 6.24 ± 0.16 | 0.90 ± 0.03 | 0.50 ± 0.03 | 25.68 |
| Sample 3 | 25.86 ± 0.74 | 2.96 ± 0.07 | 0.91 ± 0.03 | 0.30 ± 0.02 | 19.78 |
| Sample 4 | 26.53 ± 0.76 | 2.98 ± 0.07 | 0.66 ± 0.02 | 0.29 ± 0.02 | 20.07 |
| Sample 5 | 36.52 ± 1.05 | 4.97 ± 0.12 | 0.34 ± 0.01 | S < LQ [1] | 15.29 |

[1] LQ = 100 ppm.

# 3. Results

*3.1. Determination of Calorific Value*

From the 5 municipal solid waste representative samples, 5 small samples with a mass of 0.5 g were extracted and analyzed with the IKA C1/12 calorimeter (Figure 4) in accordance with the ISO 1928: 2009 standard [28]. The method for determining gross calorific values in the calorimeter IKA C1/12 is presented below. Thus, the mixed and homogenized samples were weighed with an accuracy of 0.0002 g.

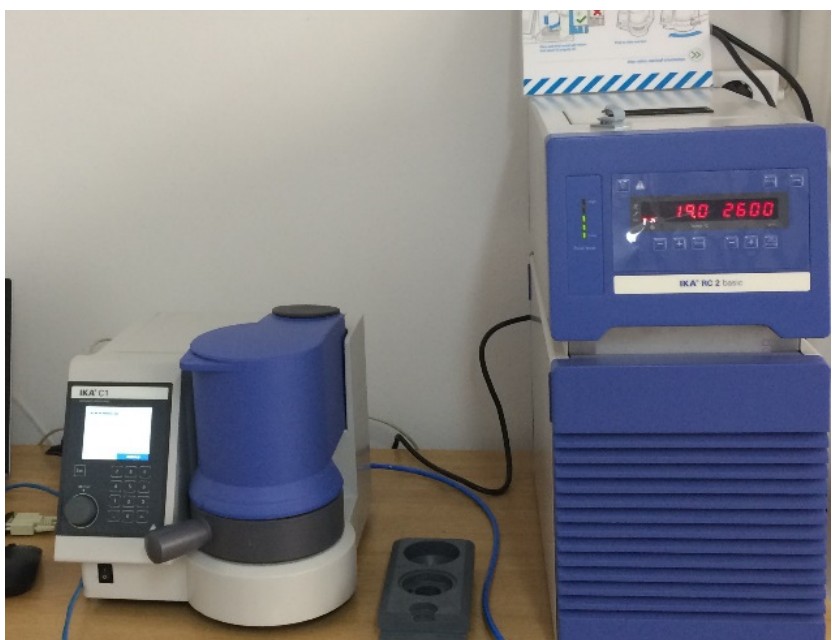

**Figure 4.** Calorimetric analysis for MSW sample.

The test sample was introduced into the decomposition vessel, where the burning under excess of oxygen in a closed container took place. The amount of heat resulting from this, measured by a previously calibrated system, allows the value of the calorific value of the sample to be determined.

The test sample is placed in a small bag, the bag in a ring-shaped crucible for burning and is closed in the decomposition vessel. A cotton thread, two electrodes and ignition wire are used for the ignition of the sample.

Five experiments were performed for each sample. The results of the experiments and of the determinations are presented in Table 5 and Figure 5.

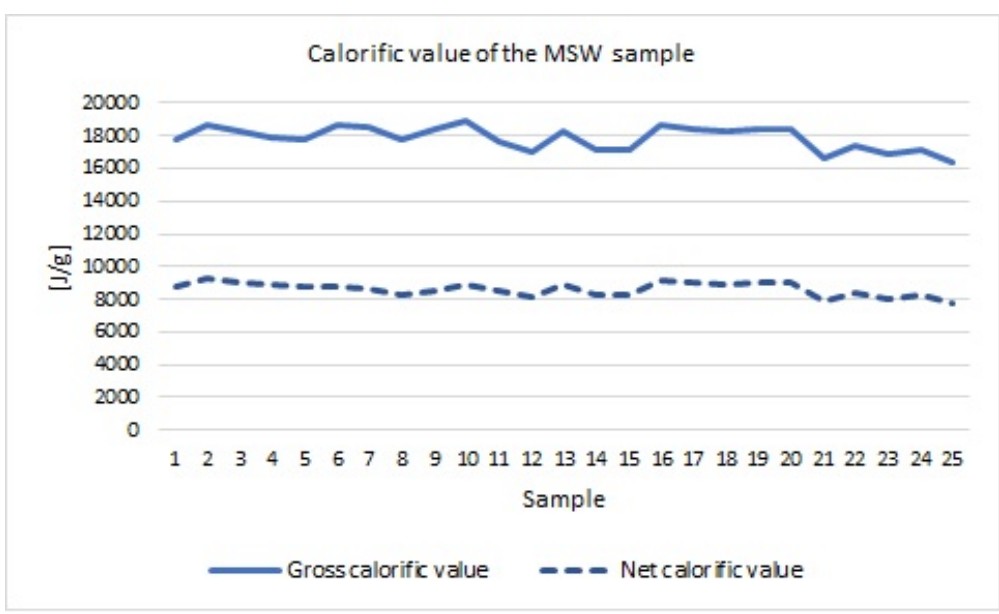

**Figure 5.** Variation of gross calorific value (GCV) and net calorific value (NCV).

**Table 5.** Calorific value of the MSW sample.

| Sample | Gross Calorific Value (GCV) $(q_{V,gr,d})$ [J/g] | Net Calorific Value (NCV) $(q_{p,net,m})$ [J/g] |
|---|---|---|
| Sample 1 | 17,822 | 8800 |
| | 18,724 | 9321 |
| | 18,264 | 9055 |
| | 17,932 | 8863 |
| | 17,825 | 8801 |
| Sample 2 | 18,633 | 8737 |
| | 18,532 | 8680 |
| | 17,828 | 8281 |
| | 18,350 | 8577 |
| | 18,958 | 8921 |
| Sample 3 | 17,636 | 8530 |
| | 17,020 | 8182 |
| | 18,283 | 8895 |
| | 17,163 | 8263 |
| | 17,089 | 8221 |
| Sample 4 | 18,643 | 9172 |
| | 18,437 | 9055 |
| | 18,254 | 8935 |
| | 18,390 | 9028 |
| | 18,389 | 9027 |
| Sample 5 | 16,676 | 7941 |
| | 17,375 | 8343 |
| | 16,893 | 8066 |
| | 17,204 | 8245 |
| | 16,422 | 7795 |
| Average | 17,869.68 | 8629.36 |
| Deviation | 711.76 | 417.37 |
| RSD (%) | 3.98 | 4.84 |

The gross calorific value (GCV) resulted from the calorimeter analysis and the net calorific value (NCV) was calculated by introducing corrections that take into account moisture, hydrogen content of the sample, oxygen content of the sample, and nitrogen content of the sample.

In order to determine the net calorific value (NCV) the following calculus formula was used, according to ISO 1928:2009 [2,28]:

$$q_{p,net,m} = \{q_{V,gr,d} - 212 \times w_{H,d} - 0.8 \times [w_{O,d} + w_{N,d}]\} \times (1 - 0.01 \times M_T) - 24.43 \times M_T \quad (2)$$

where:

- $q_{p,net,m}$ is the net calorific value of the fuel at constant pressure and water content;
- $q_{V,gr,d}$ is the gross calorific value at constant volume and free water;
- $w_{H,d}$ is the hydrogen content of the sample, expressed as a percent mass fraction, of the moisture-free (dry) fuel;
- $w_{O,d}$ is the oxygen content of the sample, expressed as a percent mass fraction, of the moisture-free fuel;
- $w_{N,d}$ is the nitrogen content of the sample, expressed as a percent mass fraction, of the moisture-free fuel.
- $M_T$ is the total moisture content, expressed as a percent mass fraction.

The resulted net calorific value (NCV) is of 8629.36 KJ/Kg.

*3.2. Energy Recovery Potential of Municipal Solid Waste*

The energy recovery potential of municipal solid waste per year can be calculated as follows [1,2,6,20]:

$$EP_{MSW/year} = W_{MSW/year} \cdot NCV / 1000 \tag{3}$$

where:

- $EP_{MSW/year}$ is the annual energy content of the treated waste, calculated on the basis of the lower net calorific value of the waste), [GJ];
- $W_{MSW/year}$-total waste quantity per year, [t];
- NCV—net calorific value, [kJ/kg].

Production of steam per year is [2]:

$$P_{steam/year} = W_{MSW/year} \times 2.5. \tag{4}$$

If the option is for waste incineration plant with cogeneration of heat and electricity can achieve an optimum energy efficiency of some 80% ($\eta C$) from total energy potential of the treated waste [8] and in this case the potential of recovered energy of MSW is:

$$EP_{CMSW} = \eta_C \times EP_{MSW}. \tag{5}$$

$EP_{CMSW}$ is calculated in MWh.

Performing the calculations according to formulas (3)–(5) the following data result:

- Energy potential of the treated waste per year $EP_{MSW}$ = 460,168 GJ = 127,824 MWh;
- production of steam per year = 133,314.7 t; and
- energy potential of electricity and thermal energy per year, $EP_{CMSW/year}$ = 102,259.58 MWh.

## 4. Discussion

The study carried out and presented in this paper is in line with the vision of the European Union which aims to introduce objectives based on the rejection of the linear economy in favor of the circular economy with the goal of recovering waste [4].

The study also aligns with Maramureș County's waste management policies and pursues the set objectives by both significantly reducing waste generation and increasing the recycling/recovery rate of waste [22].

As stated by the authors of the studies conducted for different locations, the energy potential of municipal solid waste depends on the calorific value of municipal solid waste and on the amount of municipal solid waste that is energy recovered. In addition to these parameters, the physical characteristics of the waste, the chemical composition and the moisture are also important parameters that influence the energy potential of MSW [1,2,6,11,14].

We noticed that the results obtained regarding the minimum quantity generated per year slightly exceeds the lower limit of 50,000 tons per year [8]. The MSW humidity, even though it is high, falls within the maximum limit of 50% [15], and the calorific value is higher than minimum limit of 7 MJ/kg [15].

After the development of the theoretical model and after the experimental researches, these two conditions: The minimum quantity and the humidity of municipal solid waste, represent in the opinion of the authors the main limitations related to the efficient application of the method.

This paper presents a first stage of the research regarding the Energy recovery of MSW by incineration, followed by other aspects related to the emissions resulting from incineration, to the heat treatment technologies of MSW with energy recovery and to the energy efficiency of the technologies to be approached in the future.

## 5. Conclusions

The energy potential of municipal solid waste depends on the calorific value of waste and on the amount of waste that is energy recovered. In addition to these parameters, the physical characteristics of the municipal solid waste, the chemical composition and the moisture are also important parameters that influence the heat treatment process. The analysis of the energy characteristics of municipal solid waste samples resulting in Maramures County indicates a satisfactory net calorific value (NCV): 8629.36 KJ/Kg and the moisture have the average 42.95%.

Municipal solid waste is currently generated at about 84,000 tons/year in Maramures. From this quantity approximately 53,325.87 t/year could be recovered energetically. If chosen waste incineration plant with cogeneration of heat and electricity can achieve an optimum energy efficiency of some 80% from total energy potential of the treated waste and the energy recovery per year is 102,259.58 MWh.

These findings give reason to assume that the solid waste from Maramures County that is landfilled would have satisfactory energy characteristics to be recovered for energy by incineration, but with the recommendation of a feasibility study to continue this research on a much larger scale, which in addition to increasing the number of samples and tests to take into account the different periods of the year (for the humidity of the samples).

**Author Contributions:** Conceptualization, M.U.; methodology, M.U. and N.S.U.; validation, N.S.U., P.M.; formal analysis, M.U. and J.J.; investigation, J.J. and V.M.B.; resources, M.U.; data curation, M.U. and N.S.U.; writing—original draft preparation, M.U.; writing—review and editing, N.S.U. and P.M.; supervision, M.U.; project administration, M.U.; funding acquisition, M.U., P.M., J.J., and V.M.B. All authors have read and agreed to the published version of the manuscript.

**Funding:** This research was funded under the project "Energy Recovery from Municipal Solid Waste by Thermal Conversion Technologies in Cross-border Region" HUSKROUA/1702/6.1/0015, funded by Hungary—Slovakia—Romania—Ukraine ENI CBC Programme 2014–2020—of the European Union.

**Institutional Review Board Statement:** Not applicable.

**Informed Consent Statement:** Not applicable.

**Data Availability Statement:** For more details on the project behind this work, visit https://enymsw.eu/#/home (access on 18 January 2021).

**Acknowledgments:** The authors are grateful for the financial support of this project and also would like to thank to the stakeholders, especially to Maramureș County Council, Maramureș Environmental Protection Agency, National Environmental Guard—Maramureș County Commissariat and SC DRUSAL Baia Mare who, through their support, made it possible to carry out the experiments within the study.

**Conflicts of Interest:** The authors declare no conflict of interest. The funders had no role in the design of the study; in the collection, analyses, or interpretation of data; in the writing of the manuscript, or in the decision to publish the results.

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
