# Peer review of "Research Regarding the Energy Recovery from Municipal Solid Waste in Maramures County Using Incineration"

_processes, doi:10.3390/pr9030514_

Round 1
Reviewer 1 Report
In overall article the quality of the work is certain, I have not found mistakes in the manuscript.
I think the quality of the figures can be improved, mostly the form of the graphics concerning the results and discussion (Fig.4). the introduction is quite well written, although I feel that there are too many abbreviations if possible try to reduce them. I would also recommend the authors to expand the introduction section to describe the reasoning and motivation of this application and identify their own scientific contribution and hypothesis more clearly. I would advise the authors to draw some implications (limitations, recommendations) for practical application of the proposed algorithm, as well as to outline future development of the proposed method. Mostly the form of the graphics concerning the results and discussion (Fig.4). I believe that the paper could be published.
Author Response
Dear sir
Thank you for the competent recommendations made that have contributed to increasing the scientific value of the paper.
The following changes were made:
- Was reduced the number of abbreviation
- The quality of figure 4 has been improved and figure 5, the graphical variation of the caloric value (measured and calculated) has been added
- It was specified, more clearly, in the Introduction the main motivation of the research.
- They were also specified in the discussion and conclusions chapters the main limitations of the proposed method and possibilities for future development.
Reviewer 2 Report
The paper addresses the energy recovery of municipal solid waste. It is one of the ways to remove waste from landfills and reduce environmental pollution. The presented study solves current problems and has the potential for further research.
Author Response
Dear sir
Thank you for your time and for the competent appreciations made.